# The structure of MgtE in the absence of magnesium provides new insights into channel gating

Fei Jin[1], Minxuan Sun[1], Takashi Fujii[2,3], Yurika Yamada[2], Jin Wang[4], Andrés D. Maturana[5], Miki Wada[6], Shichen Su[7], Jinbiao Ma[7], Hironori Takeda[8], Tsukasa Kusakizako[9], Atsuhiro Tomita[9], Yoshiko Nakada-Nakura[10], Kehong Liu[10], Tomoko Uemura[10], Yayoi Nomura[10], Norimichi Nomura[10], Koichi Ito[6], Osamu Nureki[9], Keiichi Namba[2,3], So Iwata[10,11], Ye Yu[4], Motoyuki Hattori[1]*

1 State Key Laboratory of Genetic Engineering, Collaborative Innovation Center of Genetics and Development, Shanghai Key Laboratory of Bioactive Small Molecules, Department of Physiology and Biophysics, School of Life Sciences, Fudan University, Shanghai, China, 2 Graduate School of Frontier Biosciences, Osaka University, Osaka, Japan, 3 Riken Quantitative Biology Center, Osaka, Japan, 4 School of Basic Medicine and Clinical Pharmacy, China Pharmaceutical University, Nanjing, China, 5 Department of Bioengineering Sciences, Graduate School of Bioagricultural Sciences, Nagoya University, Nagoya, Japan, 6 Department of Computational Biology and Medical Sciences, Graduate School of Frontier Sciences, The University of Tokyo, Chiba, Japan, 7 State Key Laboratory of Genetic Engineering, Collaborative Innovation Center of Genetics and Development, Multiscale Research Institute for Complex Systems, Department of Biochemistry, School of Life Sciences, Fudan University, Shanghai, China, 8 Faculty of Life Sciences, Kyoto Sangyo University, Kyoto, Japan, 9 Department of Biological Sciences, Graduate School of Science, The University of Tokyo, Tokyo, Japan, 10 Department of Cell Biology, Graduate School of Medicine, Kyoto University, Kyoto, Japan, 11 RIKEN SPring-8 Center, Kouto, Hyogo, Japan

* hattorim@fudan.edu.cn

**Data Availability Statement:** All raw data for the Figures presented in the manuscript can be found in the Supplementary Information. The atomic coordinates and structural factors of the MgtE-Fab structure were deposited in the Protein Data Bank.

## Abstract

MgtE is a Mg$^{2+}$ channel conserved in organisms ranging from prokaryotes to eukaryotes, including humans, and plays an important role in Mg$^{2+}$ homeostasis. The previously determined MgtE structures in the Mg$^{2+}$-bound, closed-state, and structure-based functional analyses of MgtE revealed that the binding of Mg$^{2+}$ ions to the MgtE cytoplasmic domain induces channel inactivation to maintain Mg$^{2+}$ homeostasis. There are no structures of the transmembrane (TM) domain for MgtE in Mg$^{2+}$-free conditions, and the pore-opening mechanism has thus remained unclear.

Here, we determined the cryo-electron microscopy (cryo-EM) structure of the MgtE-Fab complex in the absence of Mg$^{2+}$ ions. The Mg$^{2+}$-free MgtE TM domain structure and its comparison with the Mg$^{2+}$-bound, closed-state structure, together with functional analyses, showed the Mg$^{2+}$-dependent pore opening of MgtE on the cytoplasmic side and revealed the kink motions of the TM2 and TM5 helices at the glycine residues, which are important for channel activity. Overall, our work provides structure-based mechanistic insights into the channel gating of MgtE.

## Introduction

Mg$^{2+}$ ions are biologically essential elements involved in various physiological processes, including the catalytic action of numerous enzymes, the utilization and synthesis of ATP, and

The accession numbers for the MgtE-Fab structure are PDB: 6LBH and EMD: EMDB-0869 (https://www.ebi.ac.uk/pdbe/entry/emdb/EMDB-0869).

**Funding:** This work was supported by funding from the Ministry of Science and Technology of China (National Key R&D Program of China: 2016YFA0502800) to M.H.; funding from the National Natural Science Foundation of China (32071234) to M.H., the Innovative Research Team of High-level Local Universities in Shanghai and a key laboratory program of the Education Commission of Shanghai Municipality (ZDSYS14005); grants from Basis for Supporting Innovative Drug Discovery and Life Science Research (BINDS) from the Japan Agency of Medical Research and Development (AMED) (grant no. 19am0101079; Support No. 0451); Research on Development of New Drugs from the AMED; and Grants-in-Aid for Scientific Research from the Japan Society for the Promotion of Science (JSPS) (nos. 18K05334 and 19H00923). The funders had no role in study design, data collection and analysis, decision to publish, or preparation of the manuscript.

**Competing interests:** The authors have declared that no competing interests exist.

**Abbreviations:** CBS, cystathionine-beta-synthase; cryo-EM, cryo-electron microscopy; CTF, contrast transfer function; DDM, n-dodecyl-beta-d-maltopyranoside; DTT, dithiothreitol; ELISA, enzyme-linked immunosorbent assay; FSC, Fourier shell correlation; FSEC, fluorescence detection size exclusion chromatography; HRV3C, human rhinovirus 3C; HS-AFM, high-speed atomic force microscopy; IPTG, isopropyl-β-D-thiogalactoside; ITC, isothermal titration calorimetry; MD, molecular dynamics; PBC, periodic boundary condition; PMSF, phenylmethanesulfonyl fluoride; POPC, phosphoryl-oleoyl phosphatidylcholine; SDS-PAGE, sodium dodecyl sulfate-polyacrylamide gel electrophoresis; SEC, size exclusion chromatography; SPC, simple point charge; TM, transmembrane.

the stabilization of RNA and DNA [1–3]. Accordingly, Mg²⁺ homeostasis is a requisite for all living organisms, and Mg²⁺ channels and transporters are pivotal in Mg²⁺ homeostasis [4–7].

MgtE belongs to a MgtE/SLC41 family of Mg²⁺ channels and transporters that are ubiquitously distributed in eukaryotes, eubacteria, and archaea [8–12]. Bacterial MgtE is a Mg²⁺-selective ion channel involved in Mg²⁺ homeostasis [13,14] and is also implicated in bacterial survival after antibiotic exposure [15]. Whereas MgtE was named as a transporter when the gene was cloned, the following patch clamp analyses of MgtE from *Thermus thermophilus* by single-channel recording showed a high conductance for Mg²⁺, and the channel activity was neither pH nor Na⁺ dependent [14,16,17]. Furthermore, MgtE does not hydrolyze ATP [17]. These results are most consistent with the bacterial MgtE acting as an ion channel.

The human orthologs of MgtE, the SLC41 proteins (SLC41A1, SLC41A2, and SLC41A3), also transport Mg²⁺ ions [10,18–20], and mutations in these proteins are associated with type 2 diabetes [21], Parkinson disease [22], and nephronophthisis [23]. In contrast to bacterial MgtE as a channel, some of the SLC41 proteins (SLC41A1) are reported to be secondary active transporters driven by Na⁺ ions [24]. This situation is similar to the case with the CLC family proteins, which consist of 2 distinct subgroups of Cl⁻ channels and Cl⁻/H⁺ transporters [25–29].

The previously determined crystal structures of MgtE from *T. thermophilus* revealed the homodimeric architecture of MgtE, consisting of transmembrane (TM) and cytoplasmic domains connected via a long soluble helix, named the "plug" [30]. A high-resolution structure of the MgtE TM domain revealed the ion selectivity filter of MgtE with the fully hydrated Mg²⁺ ion in the ion-conducting pore [16]. The cytoplasmic domain of MgtE is composed of 2 subdomains, namely, the N and cystathionine-beta-synthase (CBS) domains. In the full-length MgtE structure in the presence of Mg²⁺ ions, the binding of multiple Mg²⁺ ions at the subunit interfaces of the cytoplasmic domain appears to stabilize the closed conformation of MgtE by fixing the orientation of the plug helices to close the ion-conducting pore (**Fig 1A**). The crystal structure of the Mg²⁺-free cytoplasmic domain exhibits a more relaxed conformation, where the N domain is dissociated from the CBS domain and where each subunit of the CBS domain and plug helix dimer are also separated from each other, which may unlock the closed conformation of the channel (**Fig 1B and 1C**). Consistent with this, electrophysiological recording of MgtE showed that a high concentration of cellular Mg²⁺ inhibits the channel gating of MgtE, whereas MgtE is in equilibrium between the open and closed states of the channel at a low concentration of cellular Mg²⁺ [14]. However, due to the lack of a MgtE TM domain structure in Mg²⁺-free conditions, the Mg²⁺-dependent structural changes in MgtE, particularly those in the TM domain, remain unknown, and thus, the channel-opening mechanisms remain unclear.

Here, we determined the cryo-electron microscopy (cryo-EM) structure of the MgtE-Fab complex in the absence of Mg²⁺ ions. The Mg²⁺-free MgtE structure revealed the cytoplasmic pore-opening motions of the MgtE TM domain. Structure-based functional analyses further clarified these structural changes on the cytoplasmic side, providing mechanistic insights into the channel gating of MgtE.

## Results

### Generation of the Fab antibody for MgtE

Many years of efforts to crystallize MgtE in Mg²⁺-free conditions yielded only poorly diffracting crystals. Thus, we decided to employ single-particle cryo-EM to determine the structure of MgtE in Mg²⁺-free conditions. However, it was still a challenge to obtain a high-resolution structure of full-length MgtE by cryo-EM due to its low molecular mass. Therefore, we

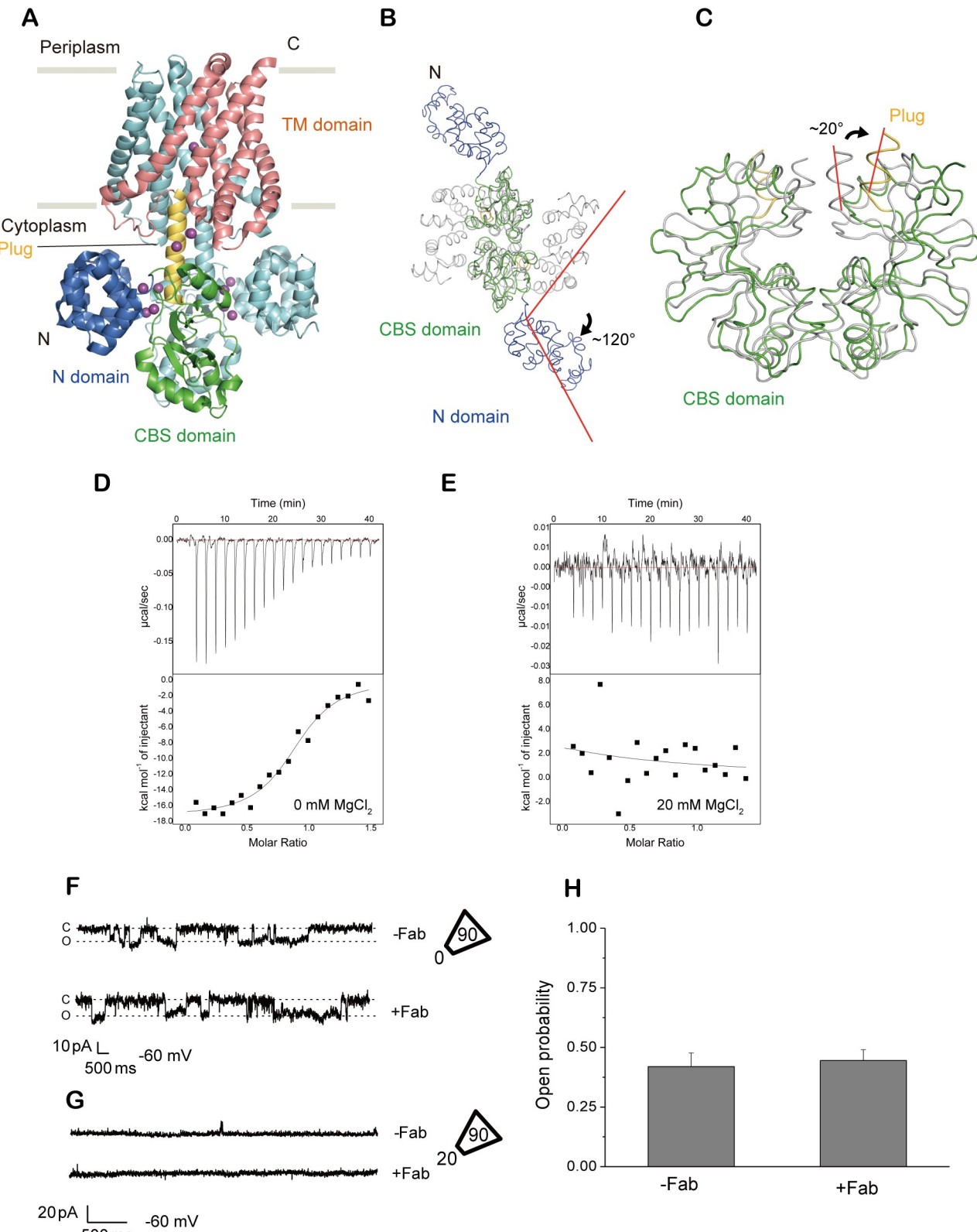

**Fig 1. Functional characterization of Fab705.** (**A**) MgtE dimer in the presence of Mg²⁺ (PDB ID:2ZY9), viewed parallel to the membrane. The N and CBS domains, plug helix and TM domain in chain A, are colored blue, green, yellow and salmon, respectively. The other subunit (chain B) is colored cyan. Mg²⁺ ions are depicted as purple spheres. (**B, C**) Structural comparison of the Mg²⁺-bound (PDB ID:2YVY) and Mg²⁺-free (PDB ID:2YVZ)

MgtE cytoplasmic domains superimposed on the CBS domains and viewed from the cytoplasmic side (**B**) and parallel to the membrane (**C**). The black arrows and red bars denote the rotation of the N domain (**B**) and plug helices (**C**). The N and CBS domains, plug helix of the Mg$^{2+}$-free state, are colored blue, green, and yellow, respectively, whereas the Mg$^{2+}$-bound state structure is colored gray. (**D, E**) Representative ITC profiles of MgtE with Fab705 in the presence of 0 mM MgCl$_2$ (D) and 20 mM MgCl$_2$ (E). (**F, G**) Single-channel recording of MgtE in the inside-out configuration from *E. coli* giant spheroplasts at −60 mV. Numbers within and outside trapezoids indicate MgCl$_2$ concentrations in pipettes and bath solutions, respectively. Representative current traces from triple-knockout *E. coli* spheroplasts expressing MgtE in the presence (lower) or absence (upper) of 2 μM Fab705. (**H**) The open probabilities for MgtE with (+) and without (−) 2 μM Fab705 ($n = 6$ for each condition, $p = 0.804$). Error bars represent the standard error of the mean. The individual numerical values that underlie the summary data displayed in this figure can be found in **S1 Data**. CBS, cystathionine-beta-synthase; ITC, isothermal titration calorimetry; TM, transmembrane.

attempted to generate Fab antibodies for MgtE to enable determination of the cryo-EM structure of the MgtE-Fab complex by increasing its total molecular mass.

After mouse immunization and generation of multiple monoclonal antibodies for MgtE, fluorescence detection size exclusion chromatography (FSEC) analysis showed that one of the Fab fragments, Fab705, bound to full-length MgtE with 0 mM Mg$^{2+}$, as indicated by the apparent peak shift (**S1A Fig**), but the binding did not occur at a high concentration of Mg$^{2+}$ (20 mM) (**S1B Fig**). Consistent with this, isothermal titration calorimetry (ITC) analysis showed that Fab705 bound to full-length MgtE with 0 mM MgCl$_2$, with a K$_d$ of 216.1 ± 52.7 nM, whereas no interaction was detected between Fab705 and MgtE at a high concentration of Mg$^{2+}$ (20 mM) (**Fig 1D and 1E, S1 Table**). We then performed patch clamp analysis of full-length MgtE using *Escherichia coli* giant spheroplasts at 0 mM MgCl$_2$ in the presence and absence of Fab705 and observed little change in the open probabilities upon addition of Fab705 into the bath solution (**Fig 1F and 1H**). Furthermore, in the patch clamp recording, when we tested a high concentration of MgCl$_2$ (20 mM MgCl$_2$) in the bath solution after the addition of Fab705, we did not observe channel opening, as similarly observed with a high concentration of MgCl$_2$ in the absence of Fab705 (**Fig 1G**). These results demonstrate that Fab705 binds to Mg$^{2+}$-free MgtE but not to MgtE with high Mg$^{2+}$ concentrations and neither positively nor negatively modulate channel opening at 0 mM Mg$^{2+}$. These functional properties of Fab705 are desirable for the determination of the MgtE structure under Mg$^{2+}$-free conditions because it does not affect channel gating under Mg$^{2+}$-free conditions, particularly in an inhibitory manner. Thus, the utilization of Fab705 enables the gating motions of MgtE to be captured by cryo-EM.

## Cryo-EM of the MgtE-Fab complex in Mg$^{2+}$-free conditions

To determine the structure of MgtE in the absence of Mg$^{2+}$ or other divalent cations, the full-length MgtE-Fab705 complex was reconstituted in PMAL-C8 amphipol and subjected to cryo-EM single-particle analysis (**Fig 2, S2 and S3 Figs**). A 3D reconstruction of the cryo-EM images was performed to reach an overall resolution of 3.7 Å, with C2 symmetry imposed (**Fig 2C**), and the cryo-EM density of the TM domain was clear enough to build an atomic model (**Fig 2D–2F, S4 Fig**). An additional cryo-EM density map of the TM domain was calculated at an approximately 3.1 Å resolution by signal subtraction and additional 3D refinement (**S5 Fig**).

Notably, we did not detect the cryo-EM density of the cytoplasmic domain of MgtE. SDS-PAGE analysis showed the presence of the cytoplasmic domain of MgtE in the purified sample for cryo-EM (**S2 Fig**), and thus, the absence of the MgtE cytoplasmic domain in the cryo-EM density was probably because of the high structural flexibility of the Mg$^{2+}$-free cytoplasmic domain [30,31]. The highly flexible nature of the Mg$^{2+}$-free MgtE cytoplasmic domain in our cryo-EM structure is consistent with previous studies conducted on MgtE by high-speed atomic force microscopy (HS-AFM) [32] and molecular dynamics (MD) simulations [31]. In particular, the HS-AFM results directly visualized the continuous shaking motions of

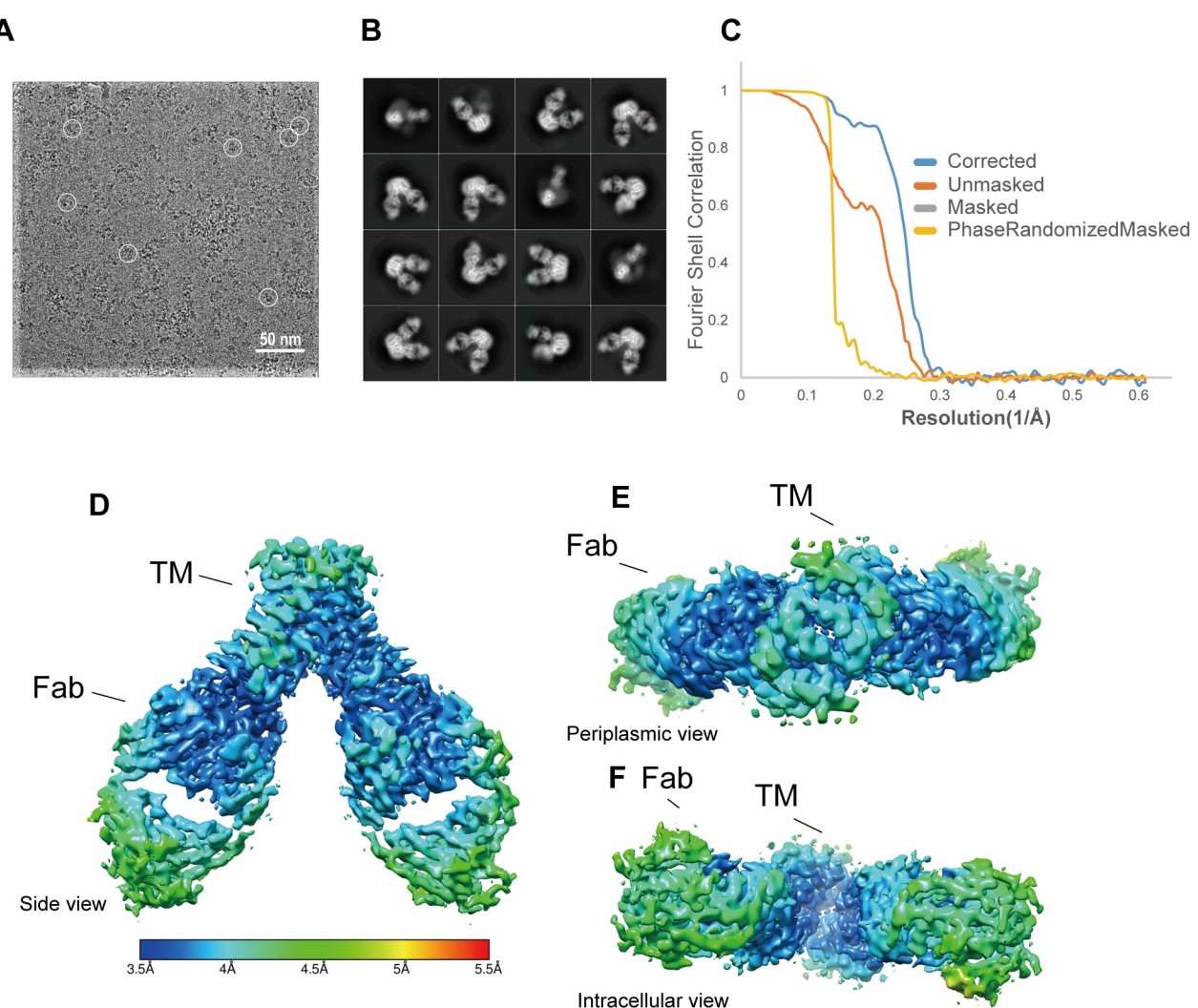

**Fig 2. Cryo-EM of the MgtE-Fab complex. (A)** Representative cryo-EM image at −1.8 μm defocused with MgtE-Fab complex particles. **(B)** Selected 2D class averages used to produce the 3D reconstruction. **(C)** Gold standard FSC for estimating resolution. **(D–F)** Side **(D)**, periplasmic **(E)**, and intracellular **(F)** views of the final map colored according to local resolution, calculated using RELION. The individual numerical values that underlie the summary data displayed in this figure can be found in **S1 Data**. cryo-EM, cryo-electron microscopy; FSC, Fourier shell correlation; TM, transmembrane.

the MgtE cytoplasmic domains under Mg$^{2+}$-free conditions. The flexibility of the MgtE cytoplasmic domain in Mg$^{2+}$-free conditions is necessary for unlocking the closed conformation of the MgtE TM domain for channel opening [14].

Nevertheless, the details of the structural changes in the TM domain following the release of Mg$^{2+}$ ions from the cytoplasmic domain were previously unclear, and for the first time, we have successfully visualized the conformation of the TM domain under Mg$^{2+}$-free conditions.

MgtE forms a symmetric dimer, and each subunit is composed of 5 TM helices (**Fig 3**). The Fab705 molecules are bound to the cytoplasmic side of the MgtE TM domain, and this binding is mediated mainly by electrostatic interactions and hydrogen bonds, primarily involving the residues in the TM2 to TM3 linker and the TM4 to TM5 linker (**Fig 3A**, **S6A Fig**). Among these residues involved in Fab binding, Arg345 interacts with Glu217 in the CBS domain, and Asp418 binds to Mg$^{2+}$ ions to bridge the TM domain and plug helices in the full-length MgtE structure in the Mg$^{2+}$-bound form (PDB ID: 2ZY9) (**S6B Fig**).

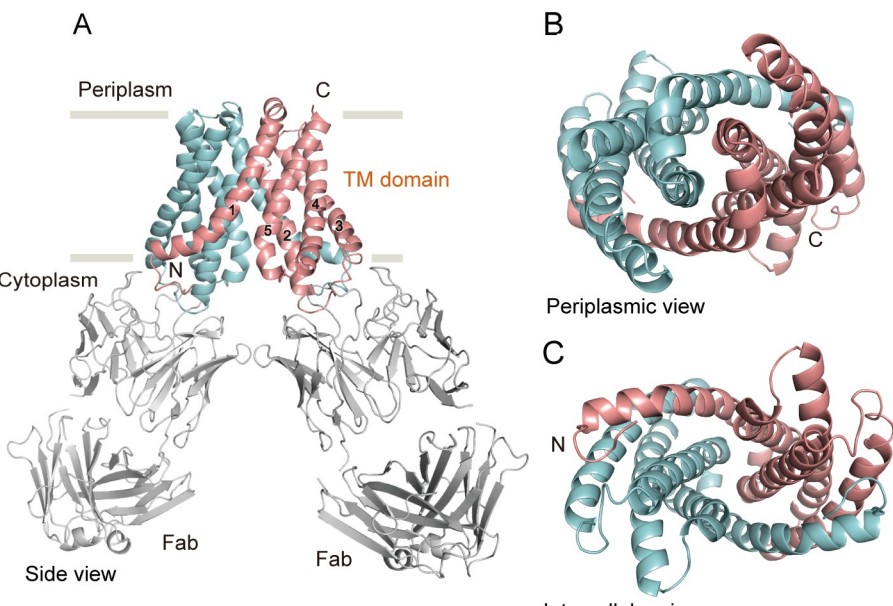

**Fig 3. Overall structure.** (**A**) Cartoon representation of the MgtE-Fab complex dimer under Mg²⁺-free conditions, viewed parallel to the membrane. The MgtE TM domain dimer is indicated in salmon (chain A) and cyan (chain B). The Fab fragments are colored in gray. (**B, C**) The MgtE dimer, viewed from the periplasm (B) and from the cytoplasm (C). TM, transmembrane.

The superposition of our cryo-EM structure and the MgtE cytoplasmic domain structure in Mg²⁺-free conditions (PDB ID: 2YVZ) onto the crystal structure of full-length MgtE in the presence of Mg²⁺ (PDB ID: 2ZY9) (**S7 Fig**) reveals the spatial collision between the Fabs and the Mg²⁺-bound cytoplasmic domain, mainly at the N domain (**S7B Fig**), whereas the N domain is dissociated from the CBS domain in the absence of Mg²⁺ (**S7B Fig**): This explains why we did not observe binding between MgtE and Fab705 at a high concentration of Mg²⁺.

Notably, in FSEC analysis, the addition of 20 mM Mg²⁺ into the sample of the MgtE-Fab complex sample, which was preformed before the addition of Mg²⁺, disrupted complex formation, where we did not see a Fab-dependent FSEC peak shift (**S1C Fig**), consistent with the patch clamp recording at a high concentration of MgCl₂ (20 mM MgCl₂) in the bath solution (**Fig 1G**). These results indicate that after Fab binding, a high concentration of Mg²⁺ ions can still close the MgtE channel by disrupting the MgtE-Fab complex.

We then compared our structure with the previously determined Mg²⁺-bound, closed-state MgtE structure (PDB ID: 2ZY9) and performed further functional analysis to gain structural insight into MgtE channel opening.

## Ion-conducting pore

The MgtE structure possesses an ion-conducting pore at the center of the dimer in the Mg²⁺-bound, closed state (**Fig 4**). The MgtE structure in Mg²⁺-free conditions has a wider pore on the cytoplasmic side than that in the previously determined structure in the presence of Mg²⁺ ions (PDB ID: 2ZY9) (**Fig 4C**). Notably, at the cytoplasmic end of the pore, the side chains of Asn424 of both subunits face each other in the Mg²⁺-bound closed structure (PDB ID: 2ZY9) (**Fig 4B**), whereas in our new Mg²⁺-free structure, the side chains of Asn424 are turned away from each other (**Fig 4A**) to form a wider cytoplasmic pore, suggesting that this region might form the cytoplasmic gate. In addition, Asn332 is in close contact with Asn424, seemingly contributing to the formation of the cytoplasmic gate (**Fig 4B**).

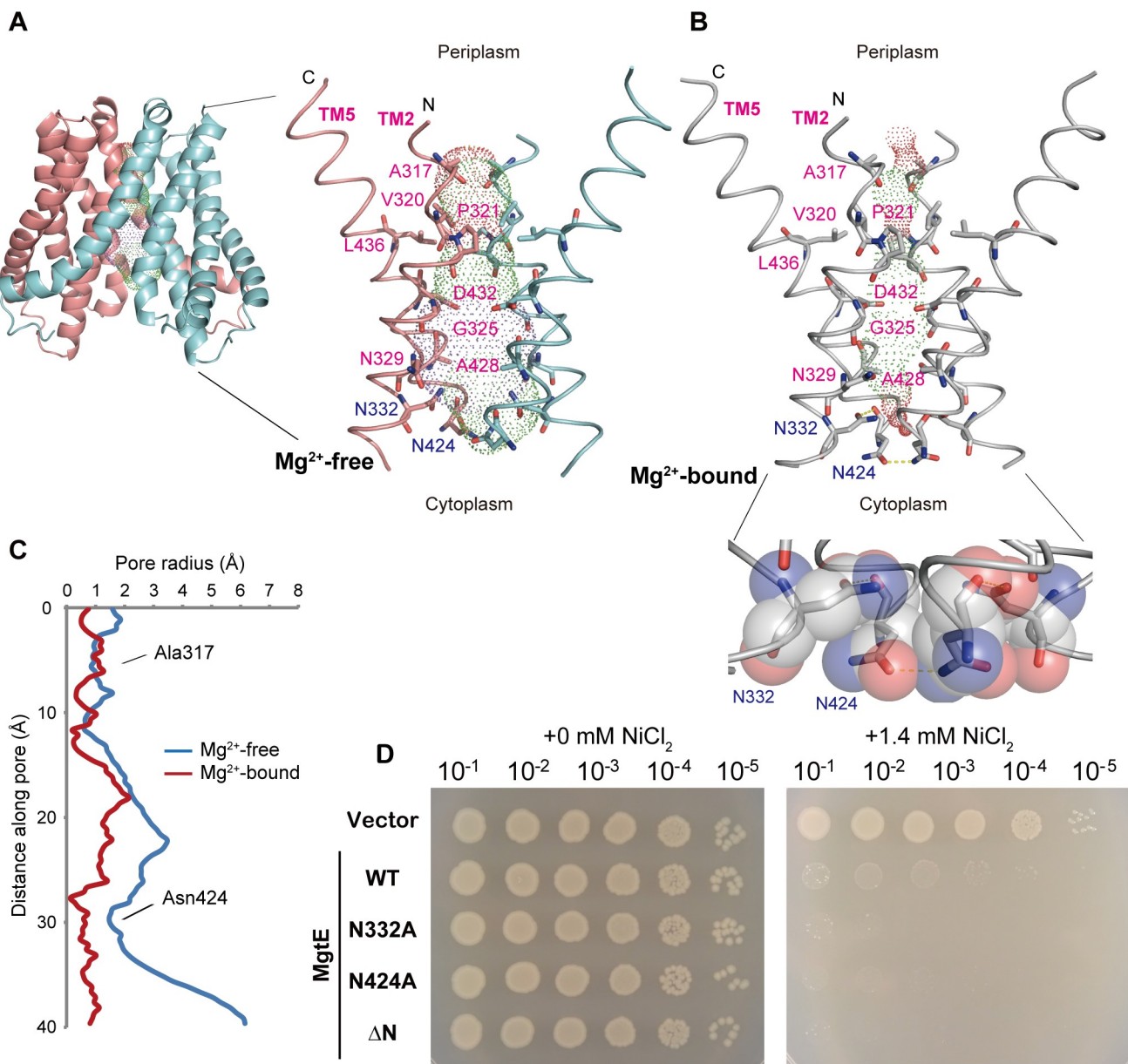

**Fig 4. Ion-conducting pore.** (**A, B**) The solvent-accessible surface of the MgtE pore under Mg$^{2+}$-free conditions (**A**) and in the presence of Mg$^{2+}$ (**B**) (PDB ID:2ZY9) with the pore-lining residues. Mg$^{2+}$-free MgtE is colored salmon (chain A) and cyan (chain B). Mg$^{2+}$-bound MgtE is colored gray. A close-up view of the cytoplasmic gate in the Mg$^{2+}$-bound state is also shown with semitransparent surface representation (**B**). Hydrogen bonds are denoted by dashed lines. A cartoon representation of the MgtE TM domain dimer under Mg$^{2+}$-free conditions is also shown (**A**). (**C**) Plots of the pore radius of MgtE structures in Mg$^{2+}$-free and Mg$^{2+}$-bound states. (**D**) Ni$^{2+}$ sensitivity assay of the WT *E. coli* strain. Serially diluted cultures ($10^{-1}$, $10^{-2}$, $10^{-3}$, $10^{-4}$, and $10^{-5}$) of magnesium-prototrophic WT *E. coli* (W3110 DE3) transformants harboring expression plasmids (empty vector, WT MgtE, N332A MgtE, N424A MgtE, and ΔN MgtE) were spotted onto the assay LB plates supplemented with 10 μM IPTG (moderate T7 promoter induction): (left) no additional NiCl$_2$, (right) + 1.4 mM NiCl$_2$. The individual numerical values that underlie the summary data displayed in this figure can be found in **S1 Data**. IPTG, isopropyl-β-D-thiogalactoside; TM, transmembrane; WT, wild-type.

To test this hypothesis, we generated MgtE mutants possessing an alanine substitution at Asn332 or Asn424, which may disrupt the cytoplasmic gate (N332A and N424A). We conducted a Ni$^{2+}$ sensitivity assay in *E. coli* using the N332A- and N424A-MgtE mutants. The coordination chemistry required for Ni$^{2+}$ is highly similar to that of Mg$^{2+}$, and MgtE and other Mg$^{2+}$ channels typically transport Ni$^{2+}$ [7]. We employed Ni$^{2+}$, not Mg$^{2+}$, for the

sensitivity assay in *E. coli* because Ni$^{2+}$ is much more toxic to *E. coli* cells than Mg$^{2+}$: Therefore, Ni$^{2+}$ is more suitable to evaluate the change in the sensitivities in *E. coli*. Increased sensitivity to Ni$^{2+}$ indicates up-regulation of Ni$^{2+}$ uptake, and the change in the sensitivities to Ni$^{2+}$ was already previously employed to characterize MgtE and its mutants in the past [17]. *E. coli* cells expressing the N332A and N424A mutants of MgtE exhibited increased Ni$^{2+}$ sensitivity (**Fig 4D**) compared to cells expressing wild-type MgtE, suggesting that the disruption of the cytoplasmic gate led to increased Ni$^{2+}$ uptake.

In contrast to the opening of the cytoplasmic side of the pore, the pore remains closed around the Ala317 residues on the periplasmic side, where the pore radius is approximately 1 Å (**Fig 4C**) in both MgtE structures in the presence and absence of Mg$^{2+}$ ions, indicating that the current structure would remain in a nonconductive conformation on the periplasmic side.

## Cytoplasmic pore-opening motions

Consistent with the wider opening of the ion-conducting pore on the cytoplasmic side, the TM helices, including the pore-forming TM2 and TM5 helices, are located in exterior positions with respect to the center of the ion-conducting pore under Mg$^{2+}$-free conditions (**Fig 5A and 5B, S1 Video**). Furthermore, the TM3 and TM4 helices have also moved away from the center, as they interact with both the TM2 and TM5 helices (**Fig 5B**). For instance, compared to the structure of the Mg$^{2+}$-bound state, Thr344 at TM2, in the Mg$^{2+}$-free state, Asp348 at TM3, Leu415 at TM4, and Ala420 at TM5 are rotated by approximately 8°, 5°, 8°, and approximately 14°, respectively, clockwise from the central axis along the pore and have moved away from the central axis by approximately 3 Å, approximately 3 Å, 3 Å, and approximately 1 Å, respectively.

To verify the Mg$^{2+}$-dependent structural changes in the TM helices on the cytoplasmic side, we generated MgtE mutants possessing cysteine substitutions at Leu421 and Thr336, where the Cα distances between Leu421 (A) and Thr336 (B) in the presence and absence of Mg$^{2+}$ ions were 5.2 and 14.0 Å, respectively (**Fig 5C**). Using Cu$^{2+}$ phenanthroline as a catalyst, cross-linking experiments were performed with these mutants (**Fig 5D and 5E**). Wild-type MgtE does not contain any endogenous cysteine residues, and the addition of MgCl$_2$ to MgtE would shift the equilibrium from the Mg$^{2+}$-free form to the Mg$^{2+}$-bound form, according to the previously reported limited proteolysis assay of MgtE [14]. The double cysteine mutant T336C/L421C exhibited a strong band corresponding to the MgtE dimer in the presence of Mg$^{2+}$ at a concentration gradient (0.01 to 30 mM) and Cu$^{2+}$ phenanthroline (**Fig 5D and 5E**), consistent with the Mg$^{2+}$-dependent structural changes observed in our cryo-EM structure.

To further support the current conformation of the MgtE TM domain structure, we performed MD simulations starting from the Mg$^{2+}$-free MgtE TM domain structure embedded in the lipid bilayer. The results of the 3-μs simulation in the absence of Mg$^{2+}$ showed that the structures of the MgtE TM domain are mostly stable (**S8A Fig**). The cytoplasmic pore remained open, with Cα distances between Leu421 (A) and Thr336 (B) of 10 to 15 Å (**S8B and S8C Fig**), whereas the periplasmic gate region site remained closed, with stable Cα distances between Ala317 (A) and Ala317 (B) of approximately 6 Å during the MD simulations (**S8D Fig**). The results further support the cytoplasmic pore opening of MgtE under Mg$^{2+}$-free conditions, as we observed in the cryo-EM structure. At the same time, we also do not exclude the possibility that we might see some structural changes if we run the MD simulation for a much longer duration.

## Kink motions of the TM helices for channel gating

Detailed observation and comparison of the MgtE structures further revealed the kink motions of TM2 at Gly325 and Gly328 by approximately 10° and of TM5 at Gly435 by approximately

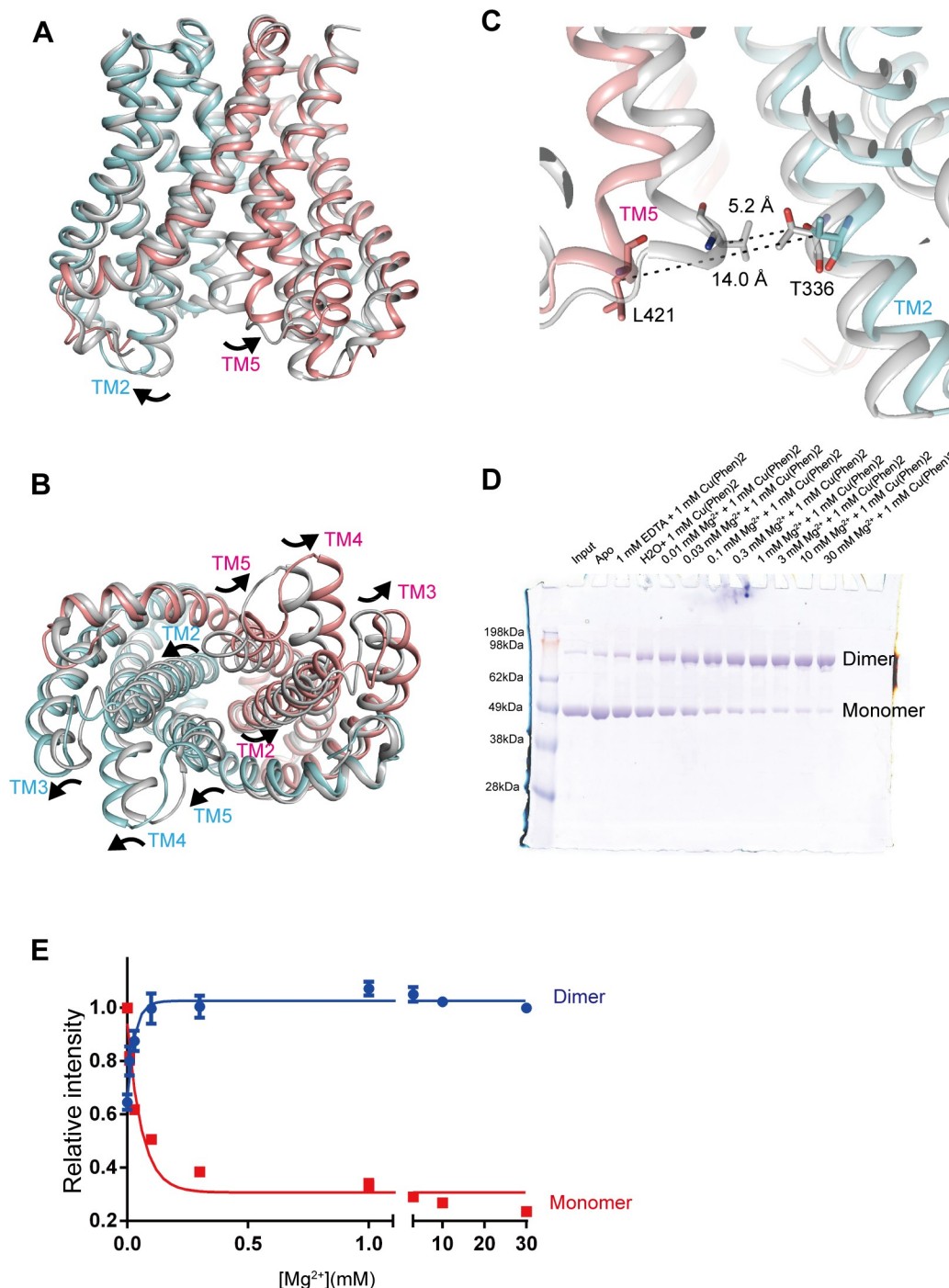

**Fig 5. Cytoplasmic pore-opening motions.** (**A, B**) Mg²⁺-free MgtE is superposed onto Mg²⁺-bound MgtE in the closed state (PDB ID:2ZY9) using the Cα positions of the TM domain dimer, viewed parallel to the membrane (**A**) and from the cytoplasm (**B**). Mg²⁺-free MgtE is colored salmon (chain A) and cyan (chain B). Mg²⁺-bound MgtE is colored gray. The black arrows indicate the structural changes between the Mg²⁺-bound and Mg²⁺-free structures. (**C**) A close-up view of the MgtE dimer interface on the cytoplasmic side. Dotted black lines indicate the Cβ distances between Thr336 and Leu421. (**D**) Representative SDS-PAGE results from chemical cross-linking experiments with the MgtE double cysteine mutant T336C/L421C. (**E**) Densitometric quantification and sigmoidal curve fitting of SDS-PAGE band intensities for the MgtE dimer (blue) and monomer (red). Experiments were repeated 6 times. Error bars represent the standard error of the mean. The individual numerical values that underlie the summary data displayed in this figure can be found in **S1 Data**. TM, transmembrane.

10° (**Fig 6A and 6B**). Since the TM2 and TM5 helices form the ion-conducting pore, these kink motions seemingly enable the TM2 and TM5 helices to expand the pore on the cytoplasmic side.

To examine the functional importance of these glycine residues, we generated a series of alanine-substituted mutants of MgtE (G325A, G328A, and G435A) and performed a growth complementation assay using a Mg²⁺-auxotrophic *E. coli* strain lacking the major genes (Δ*mgtA* Δ*corA* Δ*yhiD*) encoding Mg²⁺ transporters that survived only when the medium was supplemented with a sufficiently high concentration of Mg²⁺ [14]. The exogenously expressed MgtE could reconstitute the Mg²⁺ transport activity of this strain, and the strain transformed with the wild-type MgtE gene could survive in Mg²⁺-free LB medium [14]. While the expression of wild-type MgtE rescued Mg²⁺-auxotrophic growth, the expression of the mutations at these glycine residues almost lost Mg²⁺-auxotrophic growth complementation activity (**Fig 6C**). In particular, mutation of the strictly conserved Gly328 completely eliminated the growth complementation activity (**Fig 6C**). These results indicated that these glycine residues at the kinks of the TM2 and TM5 helices are important for the channel activity of MgtE.

## Discussion

In this work, we determined the cryo-EM structure of MgtE in Mg²⁺-free conditions. The utilization of antibodies has been successfully applied to the single-particle analysis of membrane proteins [33–37]. Notably, in our MgtE-Fab complex structure, the ordered region of MgtE (MgtE TM domain dimer) has a molecular mass of only 39 kDa, further demonstrating antibody-based techniques to be powerful tools for the determination of membrane protein structures by cryo-EM. Notably, Fab705 employed for cryo-EM does not have inhibitory effects on the channel opening under Mg²⁺-free conditions (**Fig 1F**), which would be desirable for determining the MgtE structure under Mg²⁺-free conditions (**S9 Fig**) because Fabs with inhibitory effects on channel gating may hinder structural changes in the TM domain, which would not

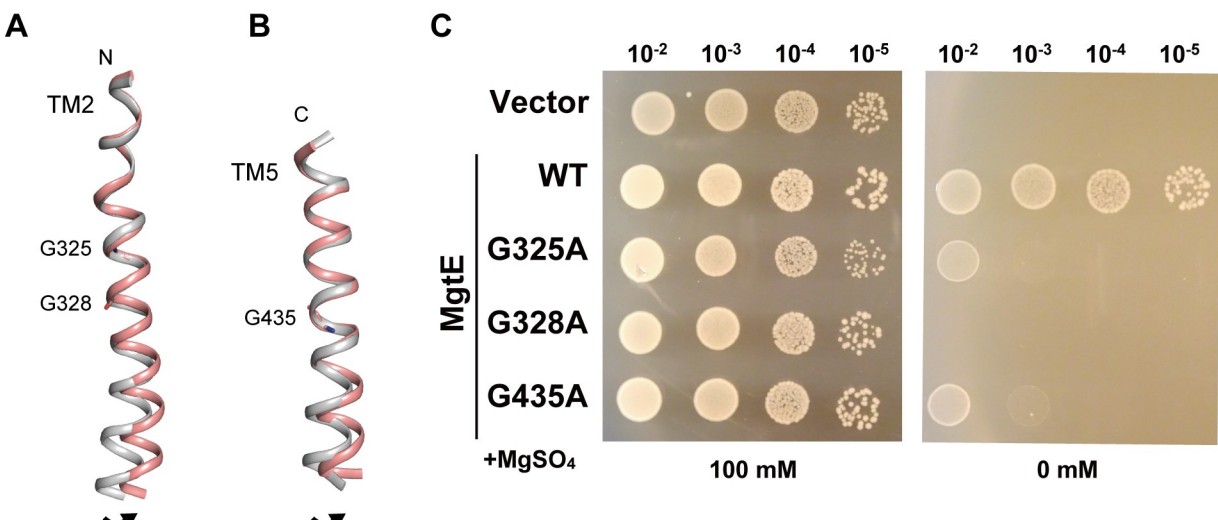

**Fig 6. Kink motions of the TM helices.** (**A, B**) Structural changes in the TM2 (A) and TM5 (B) helices. The TM helices of Mg²⁺-free MgtE (red) are superimposed on those of Mg²⁺-bound MgtE (gray) (PDB ID:2ZY9) using the Cα positions of residues 313–325 for TM2 and 435–447 for TM5. Gly325, Gly328, and Gly435 in Mg²⁺-free MgtE are depicted as stick representations. (**C**) Mg²⁺-auxotrophic growth complementation assay of MgtE and its mutants on −Mg²⁺ plates. Serially diluted cultures ($10^{-2}$, $10^{-3}$, $10^{-4}$, and $10^{-5}$) of Mg²⁺-auxotrophic *E. coli* transformants harboring expression plasmids (vector, WT MgtE, G325A MgtE, G328A MgtE, and G435A MgtE) were spotted onto the assay LB plates. TM, transmembrane; WT, wild-type.

be suitable to gain structural insights into the channel gating. Based on this structure, we further conducted structure-based functional analysis, and our new findings together with our previous research on MgtE provided structural insights into the gating mechanism of MgtE, as discussed below.

First, under high $Mg^{2+}$ concentrations, the binding of intracellular $Mg^{2+}$ to the cytoplasmic domain of MgtE, including the plug helices, stabilizes the tightly packed, closed conformation of the protein, leading to pore closure in the TM domain through extensive interactions between the plug helices and the TM domain, mainly via helices TM2 and TM5 on the cytoplasmic side [30] (**Figs 1A and 7A**).

In contrast, under $Mg^{2+}$-free conditions, the MgtE cytoplasmic domain, including the plug helices, was totally disordered and could not be built into the cryo-EM map, indicating the highly flexible nature of the MgtE cytoplasmic domain under $Mg^{2+}$-free conditions (**Fig 7B**). In such disordered conditions, the plug helices of MgtE cannot tightly interact with the TM domain to lock the channel in the closed state anymore, unlike the structure in the $Mg^{2+}$-bound state (**Figs 1A and 7A**). In other words, under $Mg^{2+}$-free conditions, the high flexibility of the cytoplasmic domain leads to the loss of the interactions between the plug helices and the TM domains, unlocking the TM domain conformation from the closed state. Highly flexible domain motions in the MgtE cytoplasmic domain were also observed by HS-AFM [32], in the crystal structure of the $Mg^{2+}$-free cytoplasmic domain structure [30], and via MD simulations of MgtE [31]. In particular, in a previous report on HS-AFM [32], Haruyama and colleagues observed a jagged topography of MgtE in the absence of $Mg^{2+}$ under HS-AFM from

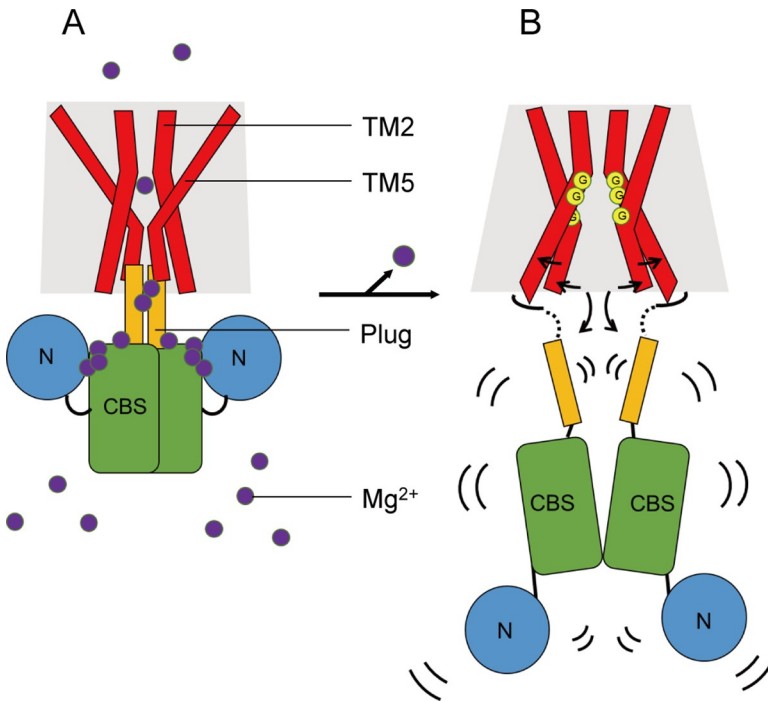

**Fig 7. Proposed mechanism.** A cartoon representation of the proposed gating motions in the presence of $Mg^{2+}$ (**A**) and under $Mg^{2+}$-free conditions (**B**). MgtE TM domain, plug helix, CBS domain, and N domain are depicted in red, yellow, green, and blue, respectively. The purple circles represent magnesium ions. The yellow circles containing the letter G emphasize the kink motions at the glycine residues (Gly325, Gly328, and Gly435). The black arrows indicate the structural changes between the $Mg^{2+}$-bound state and the $Mg^{2+}$-free structures. Curved black lines indicate the flexibility of the cytoplasmic domain under $Mg^{2+}$-free conditions. CBS, cystathionine-beta-synthase; TM, transmembrane.

the end-up orientation (cytoplasmic view), indicating that the cytoplasmic domains are constantly fluctuating. Furthermore, they observed that in the side-on orientation (side view), the cytoplasmic regions of MgtE were highly flexible and ambiguous in the absence of Mg$^{2+}$, and successive images demonstrated the association and disassembly of the N and CBS domains in solution. Such disassembly of the N and CBS domains is consistent with the previous structure of the MgtE cytoplasmic domain under Mg$^{2+}$-free conditions (**Figs 1B and 7B**). Intriguingly, similar types of regulatory mechanisms, in which the cytoplasmic domain undergoes a transition from a rigid conformation to a highly flexible or disordered state upon the release of ligands from the cytoplasmic domain, have been observed in other channels/transporters such as the CLC Cl$^−$ channel and the OpuA ABC transporter, which also possesses a CBS domain [38,39].

Following the unlocking of the TM domain due to the high flexibility of the Mg$^{2+}$-free cytoplasmic domain, we observed the structural differences of the TM domains between our cryo-EM structure and the previously determined Mg$^{2+}$-bound, inactivated structure (**Fig 5**). In our cryo-EM structure, the ion-conducting pore is wide open on the cytoplasmic side but still closed on the periplasmic side (**Fig 4**), indicating that the pore is still nonconductive. According to previous patch clamp recording results, MgtE is in equilibrium between the open and closed states under low-Mg$^{2+}$ conditions rather than in a constitutively open state [14]. Therefore, our cryo-EM structure of MgtE may represent a preopening state of MgtE under low-Mg$^{2+}$ conditions, where the pore is partially opened on the cytoplasmic side due to loss of the tight interactions with the plug helices (**Fig 7B**). Intriguingly, after the 3-μs MD simulation of the Mg$^{2+}$-free, MgtE TM domain structure was performed by adding Mg$^{2+}$ ion to the Mg$^{2+}$ binding site in the pore; the entrance region of the periplasmic side between the TM1 and TM2 helices became partially opened, with Cα distances between Glu311 (A) and Glu311 (B) of approximately 12 Å (**S8E–S8G Fig**), whereas the periplasmic gate remained nonconductive, with Cα distances between Ala317 (A) and Ala317 (B) of approximately 6 Å (**S8D Fig**). To fully understand the gating cycle of the MgtE channel, a structure in a fully opening state is still required. Furthermore, whereas the previous results are most consistent with the bacterial MgtE acting as an ion channel [14,16,17], since some of the SLC41 proteins, the eukaryotic counterparts of the bacterial MgtE, are Na$^+$-driven Mg$^{2+}$ transporters [24], the understanding of their transport mechanism would require structural analysis of eukaryotic SLC41 proteins.

In our structure, the structural changes in the TM domain occur mainly at the TM2 and TM5 helices (**Figs 5 and 7B**), and the glycine residues in the TM2 and TM5 helices enable the kink motions of the TM2 and TM5 helices for cytoplasmic pore opening (**Figs 6 and 7B**). We verified these structural changes by biochemical cross-linking experiments and genetic assays (**Figs 4D, 5D, 5E and 6C**). Notably, none of these functional experiments to verify the structural changes were performed in the presence of Fab705, suggesting that these structural changes in our MgtE-Fab complex structure are unlikely to be nonphysiological artifacts due to the use of Fab for structure determination.

Taken together, our structural and functional analyses provide mechanistic insight into the channel-opening mechanisms of MgtE, particularly cytoplasmic pore opening.

## Materials and methods

### Preparation of the MgtE-Fab complex

The full-length *T. thermophilus* MgtE gene was subcloned into a pET28a-derived vector containing an N-terminal hexahistidine tag and a human rhinovirus 3C (HRV3C) protease cleavage site, as described previously [30]. The protein was overexpressed in *E. coli* C41 (DE3) cells in LB medium containing 30 μg/ml kanamycin at 37°C by induction at an OD$_{600}$ of

approximately 0.5 with 0.5 mM isopropyl-β-D-thiogalactoside (IPTG) for 16 hours at 18°C. The cells were subsequently harvested by centrifugation (6,000 × g, 15 minutes) and resuspended in buffer H [50 mM HEPES (pH 7.0), 150 mM NaCl, 0.5 mM phenylmethanesulfonyl fluoride (PMSF)]. All purification steps were performed at 4°C. The cells were then disrupted with a microfluidizer, and the supernatants were collected after centrifugation (20,000 × g, 30 minutes) and ultracentrifugation (200,000 × g, 1 hour). Next, the membrane fraction was solubilized with buffer S [50 mM HEPES (pH 7.0), 150 mM NaCl, 2% (w/v) n-dodecyl-beta-d-maltopyranoside (DDM) (Anatrace, United States of America) and 0.5 mM PMSF] for 2 hours. The solubilized fraction was loaded onto a Ni-NTA column pre-equilibrated with buffer A [50 mM HEPES (pH 7.0), 150 mM NaCl, and 0.05% (w/v) DDM] containing 20 mM imidazole and mixed for 1 hour. The column was then washed with buffer A containing 50 mM imidazole, and the protein was eluted with buffer A containing 300 mM imidazole. The N-terminal hexahistidine tag was cleaved by HRV3C protease against dialysis buffer (buffer A containing 20 mM imidazole) overnight, and the cleaved protein was reloaded onto a Ni-NTA column pre-equilibrated with dialysis buffer.

The flow-through fraction from the column was collected and concentrated using an Amicon Ultra 50K filter (Merck Millipore, USA) and then loaded onto a Superdex 200 10/300 size exclusion chromatography (SEC) column (GE Healthcare, USA) in buffer B [20 mM HEPES (pH 7.0), and 150 mM NaCl] containing 0.05% (w/v) DDM. The main peak fractions from the column were collected and mixed with PMAL-C8 amphipol (Anatrace) at a ratio of 1:5 (w:w) overnight. The sample was mixed with Bio-Beads SM-2 (Bio-Rad, USA) and rotated gently for 4 hours. The Bio-Beads were subsequently removed by a disposable Poly-Prep column, and the eluents were concentrated using an Amicon Ultra 50K filter and loaded onto a Superdex 200 10/300 SEC column in buffer B. The SEC fractions corresponding to the amphipol-reconstituted MgtE were collected, mixed with the Fab antibody fragment Fab705 at a ratio of 1:2 (w:w) for 1 hour, and applied to a Superdex 200 10/300 SEC column in buffer B. The SEC fractions corresponding to the MgtE-Fab complex were collected and concentrated to approximately 1 mg/ml with an Amicon Ultra 50K filter for single-particle analysis using cryo-EM.

## EM data acquisition and analysis

A total of 4.0 μl of the MgtE-Fab complex in PMAL-C8 was applied to a glow-discharged holey carbon-film grid (Zhongkexinghua BioTech, GiG-A31213, 300M-Au-R1.2/1.3, China) blotted with a Vitrobot (FEI, USA) system using a 1.0-second blotting time with 100% humidity at 9°C and plunge-frozen in liquid ethane. Cryo-EM images were collected on a Titan Krios (FEI) electron microscope operated at an acceleration voltage of 300 kV. The specimen stage temperature was maintained at 80 K. Images were recorded with a K2 Summit direct electron detector camera (Gatan, USA) set to super-resolution mode with a pixel size of 0.41 Å (a physical pixel size of 0.82 Å) and a defocus ranging from −1.5 μm to −2.3 μm. The dose rate was 8 e$^-$ s$^{-1}$, and each movie was 6-second long, dose-fractioned into 40 frames with an exposure of 1.75 e$^-$ Å$^{-2}$ for each frame. The cryo-EM data are summarized in **S2 Table**.

## Image processing

A total of 5,610 movies of the MgtE-Fab complex were motion-corrected and binned with MotionCor2 [40] with 5 × 5 patches, producing summed and dose-weighted micrographs with a pixel size of 0.82 Å. Contrast transfer function (CTF) parameters were estimated by CTFFIND 4.1 [41]. Particle picking and further image processing were performed using RELION 3.0 [42]. A total of 1,860,461 particles were autopicked and extracted with a box size of 256 × 256 pixels. After several rounds of 2D classification, the re-extracted particles were

classified into 12 classes for 3D classification without any symmetry imposed. Refinement and postprocessing using RELION produced a final map at a 3.7 Å resolution from 168,911 particles with C2 symmetry imposed. The final 3D reconstruction was calculated from $2 \times 2$ binned images (0.82 Å). The final resolution was estimated using the Fourier shell correlation (FSC) = 0.143 criterion on the corrected FSC curves, in which the influence of the mask was removed. The local resolution was estimated using RELION. The workflow for image processing and for the 3D reconstruction and angular distribution plot are shown in **S3 Fig**.

## Model building

The initial model of Mg²⁺-free MgtE-Fab was manually built starting from the MgtE TM domain structure and the homology model of Fab705 generated by SWISS-MODEL [43]. Manual model building was performed using Coot [44]. Real-space refinement was performed using PHENIX software [45]. The final atomic model of MgtE includes residues 271 to 448 for chains A and B. All figures showing structures were generated using PyMol (https://pymol.org/). The sequence alignment figure was generated using Clustal Omega [46] and ESPript 3.0 [47]. The MgtE pore shown in **Fig 4** was calculated by HOLE [48].

## Preparation of antibody fragments

Mouse monoclonal antibodies against MgtE were raised by an established method [49]. Briefly, a proteoliposome antigen was prepared by reconstituting purified, functional MgtE at high density in phospholipid vesicles consisting of a 10:1 mixture of egg phosphatidylcholine (Avanti Polar Lipids, USA) and the adjuvant lipid A (Sigma-Aldrich, USA) to facilitate the immune response. BALB/c mice immunization was performed with the proteoliposome antigen using 3 injections at 2-week intervals. Antibody-producing hybridoma cell lines were generated using a conventional fusion protocol [50]. Hybridoma clones producing antibodies recognizing conformational epitopes in MgtE were selected by an enzyme-linked immunosorbent assay on immobilized phospholipid vesicles containing purified MgtE (liposome ELISA), allowing positive selection of the antibodies that recognized the native conformation of MgtE. Additional screening for reduced antibody binding to SDS-denatured MgtE was used for negative selection against linear epitope-recognizing antibodies. Stable complex formation between MgtE and each antibody clone was checked using FSEC. From these screens, we isolated 4 MgtE-specific monoclonal antibodies: YN0705, YN0710, YN0721, and YN0736. Whole IgG molecules, collected from the large-scale culture supernatant of monoclonal hybridomas and purified using protein G affinity chromatography, were digested with papain, and Fab fragments were isolated using a HiLoad 16/600 Superdex 200 gel filtration column followed by protein A affinity chromatography. The cDNAs encoding the light and heavy chains of Fab YN0705 were cloned from hybridoma cells using rapid amplification of 5′ complementary DNA ends (5′-RACE) and sequenced.

## Isothermal titration calorimetry

The binding affinities of the Fab antibody fragment Fab705 for full-length MgtE were measured using MicroCal iTC200 (GE Healthcare). The full-length MgtE proteins were purified by a similar procedure to that described above but with SEC buffer [buffer B containing 0.05% (w/v) DDM] containing either 0 or 20 mM MgCl₂. Fab was also used for gel filtration with the same SEC buffer containing either 0 or 20 mM MgCl₂. The peak fractions of MgtE were collected and diluted to approximately 2.5 μM (monomer) with SEC buffer, while Fab was diluted to 25 μM. Fab was injected into the cell containing the full-length MgtE 20 times (0.5 μl for injection 1, 2 μl for injections 2 to 20) at 25˚C, with 120-second intervals between injections.

The background data were measured by injecting the SEC buffer into a cell containing the full-length MgtE. The results were analyzed by Origin8 software (MicroCal). The experiments were repeated 3 times for each measurement, and similar results were achieved. All ITC profiles are shown in **S10 Fig**. The protein concentration was estimated using the absorbance at 280 nm.

## Chemical cross-linking

The wild-type and mutant MgtE proteins were purified as described above, except that dithiothreitol (DTT) was added at a final concentration of 20 mM to the MgtE mutant proteins before removal by SEC, and then diluted to 0.5 mg/ml with buffer B containing 0.05% (w/v) DDM. MgtE proteins (4.0 μl) were mixed with 0.5 μl of MgCl₂ at appropriate concentrations and incubated for 30 minutes at 4°C. Then, 0.5 μl of 10 mM Cu²⁺ bis-1,10-phenanthroline was added at a 1:3 molar ratio, and the samples were incubated for another 30 minutes at 4°C. The reaction mixtures were loaded with nonreduced loading buffer and analyzed immediately by sodium dodecyl sulfate-polyacrylamide gel electrophoresis (SDS-PAGE). The relative intensity of the bands was analyzed by ImageJ software and fitted to a sigmoidal equation. The experiments were repeated 6 times. All SDS-PAGE results are shown in **S11 Fig**.

## Molecular dynamics simulations

As described previously [51,52], DESMOND [53] was used to perform MD analysis with a constant number of particles, constant pressure (1 bar) and temperature (300 K) (NPT), and periodic boundary conditions (PBCs) by using the Nose–Hoover chain thermostat and Martyna–Tobias–Klein barostat methods. The MgtE TM domain dimer was embedded in a phosphoryl-oleoyl phosphatidylcholine (POPC) bilayer. The structure of MgtE/POPC was solvated in an orthorhombic box of SPC water molecules. All-atom OPLS_2005 force fields for proteins, ions, lipids, and simple point charge (SPC) waters were used in all simulations [54,55]. Before performing simulations, a default relaxation protocol in DESMOND was employed: (1) NVT ensemble with Brownian dynamics for 100 ps at 10 K with small time step and solute heavy atom restrained; (2) NVT ensemble using Berendsen thermostat for 12 ps at 10 K with small time step and solute heavy atom restrained; (3) NPT ensemble using a Berendsen thermostat and barostat for 12 ps at 10 K and 1 atm with solute heavy atom restrained; (4) NPT ensemble using a Berendsen for 12 ps at 300 K and 1 atm with solute heavy atom restrained; and (5) NPT ensemble using a Berendsen for 24 ps at 300 K and 1 atm with no restraints. After the relaxation, MD simulation was performed for 200 ns. The integration time step used was 2 fs, and coordinate trajectories were saved every 200 ps. A DELL T7910 graphics workstation (with multiple NVIDIA Tesla K40C GPUs) was used to run the MD simulations, and a 12-CPU CORE DELL T7500 graphics workstation was used to perform the preparation, analysis, and visualization. The Simulation Event Analysis module in DESMOND was used to analyze MD trajectories.

## Electrophysiological recordings

Patch clamp recording of MgtE using *E. coli* giant spheroplasts was performed in the inside-out configuration, as described previously [14,56]. In brief, spheroplasts were placed in a bath solution containing 210 mM N-methyl-D-glucamine, 90 mM MgCl₂, 300 mM D-glucose, and 5 mM HEPES (pH 7.2). After gigaseal formation with a borosilicate glass pipette (Harvard Apparatus, Kent, United Kingdom) with 6 to 8 mOhm resistance, the bath medium was changed to bath solution containing 300 mM N-methyl-D-glucamine, 300 mM D-glucose, and 5 mM HEPES (pH 7.2) by perfusion using a Rainer perfusion pump and a custom-made

perfusion system. Bath medium containing 2 μM Fab705 was added by perfusion. The pipette solution contained 210 mM N-methyl-D-glucamine, 90 mM $MgCl_2$, 300 mM sucrose, and 5 mM HEPES (pH 7.2). Currents were measured with an Axopatch 200B amplifier and recorded with a Digidata 1440B A/D converter under the control of pCLAMP software (Molecular Devices, USA). Currents were filtered at 2 kHz and recorded at 5 kHz. Additional reporting details regarding the electrophysiology experiments are shown in S12 Fig.

## *E. coli* genetics

As described previously [14], the $Mg^{2+}$-auxotrophic *E. coli* strain (BW25113 Δ*mgtA* Δ*corA* Δ*yhiD* DE3) was transformed with MgtE or MgtE mutant plasmids and grown on LB medium plates containing 50 μg/ml kanamycin and 100 mM $MgSO_4$. Each transformant was inoculated in liquid LB medium containing 50 μg/ml kanamycin and 100 mM $MgSO_4$ and incubated at 37°C overnight with shaking. The liquid samples were diluted 1:100 with the same medium and grown until the OD600 reached 0.6. The sample cultures were then serially diluted 10-fold in LB medium containing 50 μg/ml kanamycin, spotted onto assay plates, and incubated at 37°C overnight, as indicated in Fig 6C.

In the $Ni^{2+}$ sensitivity assay, the isogenic magnesium-prototrophic wild-type *E. coli* strain (W3110 DE3) was transformed with the plasmid, and transformants were obtained on LB (+50 μg/ml kanamycin) plates. The liquid samples were diluted 1:100 with the same medium and grown until the OD600 reached 0.6. The sample cultures were then serially diluted 10-fold and spotted onto assay plates, and the plates were incubated at 37°C overnight, as indicated in Fig 4D.

The expression of MgtE and its mutants was confirmed by western blotting (S13 **and** S14 **Figs).**

## Western blot analysis

The $Mg^{2+}$-auxotrophic *E. coli* strain (BW25113 Δ*mgtA* Δ*corA* Δ*yhiD* DE3) was transformed with the plasmids, and transformants were obtained on LB (+50 μg/ml kanamycin) plates supplemented with 100 mM $MgSO_4$. Each transformant was cultured in LB liquid medium supplemented with 50 μg/ml kanamycin and 100 mM $MgSO_4$ overnight. The *E. coli* cell numbers were adjusted based on the OD600. Whole-cell extracts were prepared, resolved by SDS-PAGE (10% polyacrylamide), and transferred to Hybond-ECL (GE Healthcare). The transferred proteins stained with an anti-His6 polyclonal antibody, anti-His-tag-HRP direct T (MBL, Japan). Antibody-bound proteins were visualized using western blot detection reagents (Immunostar LD, Wako Laboratory Chemicals, Japan). Images were captured with an LAS-3000 Mini imaging system (Fujifilm, Japan).

## FSEC experiments

Purified full-length MgtE and Fab705 proteins were diluted to 1 mg/ml with buffer B containing 0.05% (w/v) DDM. The MgtE-Fab complex was formed by mixing MgtE and Fab at mass ratios of 1:0, 0:1, 1:0.5, 1:1, and 1:2 in the presence of 0 mM or 20 mM $Mg^{2+}$. The samples were then loaded onto a Superdex 200 Increase 10/300 SEC column (GE Healthcare) connected to an RF-20Axs fluorescence detector (Shimadzu, Japan) (excitation: 280 nm, emission: 325 nm) with buffer B containing 0.05% (w/v) DDM and 0 mM or 20 mM $Mg^{2+}$.

The complex disruption experiment was performed by adding $Mg^{2+}$ at a final concentration of 20 mM to the preformed MgtE-Fab complex, which was prepared by mixing MgtE and Fab705 at a mass ratio of 1:2 in 0 mM $Mg^{2+}$ and loading the sample on an FSEC column with buffer B containing 0.05% (w/v) DDM and 20 mM $Mg^{2+}$.

## Supporting information

**S1 Fig. FSEC analysis of MgtE-Fab complex formation.** (**A, B**) FSEC analysis of MgtE-Fab complex formation. The MgtE-Fab complex was formed by adding Fab to MgtE in 0 mM $MgCl_2$ (**A**) and in 20 mM $MgCl_2$ (**B**) at the indicated mass ratios. The FSEC running buffer included 0 mM for (**A**) and 20 mM $MgCl_2$ for (**B**). (**C**) The complex disruption experiment was performed by adding $Mg^{2+}$ at a final concentration of 20 mM to the preformed MgtE-Fab complex, which was prepared by mixing MgtE and Fab705 at a mass ratio of 1:2 in 0 mM $Mg^{2+}$. In addition to the FSEC profile for the complex disrupted sample (blue), the FSEC profiles of Fab-free MgtE in $Mg^{2+}$-free conditions (orange) and 20 mM $MgCl_2$ (gray) are also shown. The FSEC running buffer includes 20 mM $MgCl_2$, except for the FSEC of MgtE in $Mg^{2+}$-free conditions. In FSEC profiles, $Mg^{2+}$-free MgtE (**A, C**) eluted earlier than $Mg^{2+}$-bound MgtE (**B, C**), probably because the conformation of MgtE, in particular the conformation of the cytoplasmic domain of MgtE, is more compact in the presence of $Mg^{2+}$ than in the absence of $Mg^{2+}$ as shown in **Fig 1B**. FSEC, fluorescence detection size exclusion chromatography. (TIF)

**S2 Fig. SEC and SDS-PAGE of the Fab-MgtE complex.** (**A**) SEC of the MgtE-Fab complex. The former, middle, and latter peaks were the void, Fab-MgtE complex, and free Fab, respectively. The fractions from the 10.5 to 12.0 ml elution positions were pooled as the cryo-EM sample. (**B**) SDS-PAGE of the SEC fractions and the cryo-EM sample. The individual numerical values that underlie the summary data displayed in this figure can be found in **S1 Data**. cryo-EM, cryo-electron microscopy; SDS-PAGE, sodium dodecyl sulfate-polyacrylamide gel electrophoresis; SEC, size exclusion chromatography. (TIF)

**S3 Fig. Flowchart for cryo-EM data processing.** (**A**) Overview of the data processing workflow, including particle picking, classification, and 3D refinement. All processing steps were performed in RELION. (**B**) Euler angle distribution plot of all particles included in the calculation of the MgtE-Fab complex, with C2 symmetry imposed. cryo-EM, cryo-electron microscopy. (TIF)

**S4 Fig. EM density map.** (**A–E**) Representative EM density maps contoured at 4.0 σ. The TM1 helix (residues 278–311) (**A**), TM2 helix (residues 315–345) (**B**), TM3 helix (residues 353–380) (**C**), TM4 helix (residues 384–416) (**D**), and TM5 helix (residues 419–448) (**E**) in chain A are shown as stick representations. EM, electron microscopy; TM, transmembrane. (TIF)

**S5 Fig. EM map for the TM domain obtained by signal subtraction.** (**A**) Side view of the EM density map for the TM domain obtained by signal subtraction, colored according to local resolution, calculated using RELION. (**B**) Gold standard FSC for estimating resolution. The individual numerical values that underlie the summary data displayed in this figure can be found in **S1 Data**. EM, electron microscopy; FSC, Fourier shell correlation; TM, transmembrane. (TIF)

**S6 Fig. Fab binding site.** A close-up view of the MgtE-Fab interface on the cytoplasmic side (**A**) and of the corresponding region in full-length MgtE in the $Mg^{2+}$-bound form (PDB ID: 2ZY9) (**B**). The coloring scheme of MgtE is the same as that in **Fig 1**, and Fab is colored gray. Residues located at the MgtE-Fab interface are depicted as stick representations. $Mg^{2+}$ ions are depicted as purple spheres. (TIF)

**S7 Fig. Structural comparisons.** Superimpositions of our cryo-EM structure of the MgtE-Fab complex under Mg$^{2+}$-free conditions and the Mg$^{2+}$-free cytoplasmic domain structure (PDB ID: 2YVZ) onto full-length MgtE in the Mg$^{2+}$-bound form (PDB ID: 2ZY9), viewed parallel to the membrane (**A**) and from the cytoplasmic side (**B**). The coloring scheme of the full-length MgtE structure is the same as that in **Fig 1**. The Mg$^{2+}$-free cytoplasmic domain structure is colored gray. The TM domain and Fabs in the MgtE-Fab complex are colored red and orange, respectively. cryo-EM, cryo-electron microscopy; TM, transmembrane.
(TIF)

**S8 Fig. MD simulations.** (**A**) Structural deviations from the Mg$^{2+}$-free MgtE TM domain structure during the 3-μs MD simulations. (**B–E**) Cα distances between Thr336 (chain B) and Leu421 (chain A) (**B**), between Thr336 (chain A) and Leu421 (chain B) (**C**), between Ala317 (chains A and B) (D), and between Glu311 (chains A and B) (**E**) during the 3-μs MD simulation. The MD simulations were performed with (red) and without (black) adding Mg$^{2+}$ ion at the Mg$^{2+}$ binding site in the pore. The position of the added Mg$^{2+}$ ion in the pore was based on the previous Mg$^{2+}$-bound MgtE structure (PDB ID: 2ZY9) (**Fig 1A**). (**F, G**) The Mg$^{2+}$-free MgtE TM domain structures after the 3-μs MD simulations with (**G**) and without (**F**) adding Mg$^{2+}$ ion in the pore are superposed onto Mg$^{2+}$-bound MgtE in the closed state (PDB ID:2ZY9) using the Cα positions of the TM domain dimer. The MgtE structures after the 3-μs MD simulations are colored salmon (chain A) and cyan (chain B). Mg$^{2+}$-bound MgtE is colored gray. Mg$^{2+}$ ions are shown as purple spheres. The black arrows indicate the structural changes. The individual numerical values that underlie the summary data displayed in this figure can be found in **S1 Data**. MD, molecular dynamics; TM, transmembrane.
(TIF)

**S9 Fig. Cartoon diagram of Fab705 effect.** A cartoon of the proposed effect of Fab705 in the presence of Mg$^{2+}$ (**A**) and under Mg$^{2+}$-free conditions (**B, C**). Briefly, Fab705 cannot bind to MgtE at high Mg$^{2+}$ concentrations (**A**). At low Mg$^{2+}$ concentrations, Fab705 can bind to MgtE but does not either positively or negatively modulate channel opening, and a high concentration of Mg$^{2+}$ ions can still close the MgtE channel by disrupting the MgtE-Fab complex (**C**).
(TIF)

**S10 Fig. ITC profiles of MgtE with Fab705.** (**A, B**) ITC profiles of MgtE with Fab705 in the presence of 0 mM MgCl$_2$ (A) and 20 mM MgCl$_2$ (B). The individual numerical values that underlie the summary data displayed in this figure can be found in **S1 Data**. ITC, isothermal titration calorimetry.
(TIF)

**S11 Fig. SDS-PAGE gels from chemical cross-linking experiments with the MgtE double cysteine mutant T336C/L421C.** SDS-PAGE, sodium dodecyl sulfate-polyacrylamide gel electrophoresis.
(TIF)

**S12 Fig. Patch clamp recording of MgtE using *E. coli* giant spheroplasts.** (**A**) Representative current traces from the same patch of triple-knockout *E. coli* spheroplasts expressing MgtE under 0 mM MgCl$_2$ (left) and under 20 mM MgCl$_2$ (right) in the bath solution. Application of the bath solution containing 20 mM MgCl$_2$ inhibited channel opening. Experiments were repeated 3 times, and similar results were obtained. (**B**) Representative current traces of triple-knockout *E. coli* spheroplasts expressing MgtE with multiple channel openings in the patch. Of the 18 patches, 14 of them showed channel currents, and 4 contained 2–4 channels. (**C**)

Representative current traces from triple-knockout *E. coli* spheroplasts harboring the empty vector. Experiments were repeated 7 times, and no channel currents were obtained. The individual numerical values that underlie the summary data displayed in this figure can be found in **S1 Data**.
(TIF)

**S13 Fig. Western blot of the MgtE N332A and N424A mutants.**
(TIF)

**S14 Fig. Western blot of the MgtE glycine mutants.** Uncropped western blots are provided in S1 Raw Images.
(TIF)

**S1 Video. Structural changes in the MgtE TM domain.** TM, transmembrane.
(MP4)

**S1 Table. ITC statistics.** ITC, isothermal titration calorimetry.
(PDF)

**S2 Table. Cryo-EM data collection, refinement, and validation statistics.** cryo-EM, cryo-electron microscopy.
(PDF)

**S1 Data. Numerical raw data.** All numerical raw data associated with **Figs 1D–1H, 2C, 4C, and 5E** and **S2A, S5B, S8A–S8E, S10A, S10B, and S12A–S12C Figs**. File contains multiple tabs with labels corresponding to the relevant figure.
(XLSX)

**S1 Raw Images. Uncropped western blots from S14 Fig.**
(TIF)

## Acknowledgments

We thank the staff scientists at the Center for Biological Imaging, Institute of Biophysics, and National Center for Protein Science Shanghai (Chinese Academy of Sciences) for technical support with cryo-EM data collection (project numbers: CBIapp20180107, CBIapp201807006, CBIapp201807007, 2017-NFPS-PT-001632, and 2018-NFPS-PT-002187); Yumi Sato for technical assistance in the generation of antibodies; Namba Laboratory members for technical support with cryo-EM data collection; Dr. Hideaki E. Kato (University of Tokyo) for critical comments on the manuscript; and Drs. Chia-Hsueh Lee (St. Jude Children's Research Hospital) and Muneyoshi Ichikawa (NAIST) for their technical advice on cryo-EM data processing.

## Author Contributions

**Conceptualization:** Motoyuki Hattori.

**Data curation:** Fei Jin, Minxuan Sun, Takashi Fujii, Yurika Yamada, Jin Wang, Andrés D. Maturana, Miki Wada, Tsukasa Kusakizako, Yoshiko Nakada-Nakura, Kehong Liu, Tomoko Uemura, Yayoi Nomura, Norimichi Nomura, Koichi Ito, Ye Yu, Motoyuki Hattori.

**Formal analysis:** Fei Jin, Minxuan Sun, Takashi Fujii, Yurika Yamada, Jin Wang, Andrés D. Maturana, Miki Wada, Yoshiko Nakada-Nakura, Yayoi Nomura, Norimichi Nomura, Koichi Ito, So Iwata, Ye Yu, Motoyuki Hattori.

**Funding acquisition:** Motoyuki Hattori.

**Investigation:** Motoyuki Hattori.

**Methodology:** Shichen Su, Jinbiao Ma, Hironori Takeda, Atsuhiro Tomita, Kehong Liu, Tomoko Uemura, Yayoi Nomura, Norimichi Nomura, Osamu Nureki, Keiichi Namba, So Iwata.

**Project administration:** Motoyuki Hattori.

**Supervision:** Motoyuki Hattori.

**Writing – original draft:** Fei Jin, Koichi Ito, Ye Yu, Motoyuki Hattori.

**Writing – review & editing:** Minxuan Sun, Takashi Fujii, Yurika Yamada, Jin Wang, Andrés D. Maturana, Miki Wada, Shichen Su, Jinbiao Ma, Hironori Takeda, Tsukasa Kusakizako, Atsuhiro Tomita, Yoshiko Nakada-Nakura, Kehong Liu, Tomoko Uemura, Yayoi Nomura, Norimichi Nomura, Osamu Nureki, Keiichi Namba, So Iwata, Motoyuki Hattori.

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
