## [Editor Report · Decision Letter 0]

25 Nov 2020

Dear Dr Hattori, 

Thank you for submitting your manuscript entitled "Cryo-EM structure of the MgtE Mg2+ channel pore domain in Mg2+-free conditions reveals cytoplasmic pore opening" for consideration as a Research Article by PLOS Biology.

Your manuscript has now been evaluated by the PLOS Biology editorial staff as well as by an academic editor with relevant expertise and I am writing to let you know that we would like to pursue your manuscript further and invite you to submit a revised version of the manuscript.

However, before we can invite a revision, we need you to complete your submission by providing the metadata that is required for full assessment. To this end, please login to Editorial Manager where you will find the paper in the 'Submissions Needing Revisions' folder on your homepage. Please click 'Revise Submission' from the Action Links and complete all additional questions in the submission questionnaire.

Please re-submit your manuscript within two working days, i.e. by Nov 27 2020 11:59PM.

Kind regards,

Richard Hodge, PhD

Associate Editor

PLOS Biology

---

## [Editor Report · Decision Letter 1]

26 Nov 2020

Dear Dr Hattori,

Thank you very much for submitting your manuscript "Cryo-EM structure of the MgtE Mg2+ channel pore domain in Mg2+ -free conditions reveals cytoplasmic pore opening" for consideration as a Research Article at PLOS Biology.

As you know, your manuscript and plan of revision have been evaluated by the PLOS Biology editors and by an Academic Editor with relevant expertise.

Based on your responses to the reviews from Reviews Commons, we would welcome re-submission of a revised version that takes into account the reviewers' comments. In addition, the Academic Editor would like you to provide a clear response to Reviewer #2 comments in the manuscript that clarifies the function of MgtE as an ion channel or a secondary active transporter.

In addition, the Academic Editor requests that you provide additional reporting details regarding the electrophysiology experiments conducted in the manuscript. We ask whether you have recorded macroscopic currents and whether a similar channel behaviour is seen. We would also like you to clarify in the manuscript whether you have recorded current-voltage relationships in asymmetric Mg2+ concentrations and assess Mg2+ selectivity based on the analysis of reversal potentials. In both cases, it would be acceptable to discuss and refer to this data by citing previous publications.

We cannot make any decision about publication until we have seen the revised manuscript and your response to the reviewers' comments. Your revised manuscript is also likely to be sent for further evaluation by the original reviewers.

We expect to receive your revised manuscript within 3 months. Please email us (plosbiology@plos.org) if you have any questions or concerns, or would like to request an extension. At this stage, your manuscript remains formally under active consideration at our journal; please notify us by email if you do not intend to submit a revision so that we may end consideration of the manuscript at PLOS Biology.

*Re-submission Checklist*

*Published Peer Review*

*PLOS Data Policy*

*Blot and Gel Data Policy*

Sincerely,

Richard Hodge, PhD

Associate Editor

PLOS Biology

---

## [Decision Letter · Decision Letter 2]

24 Mar 2021

Dear Dr Hattori,

Thank you for submitting your revised Research Article entitled "Cryo-EM structure of the MgtE Mg2+ channel pore domain:Fab complex in Mg2+-free conditions reveals cytoplasmic pore opening" for publication in PLOS Biology. I have now obtained advice from the original reviewers and have discussed their comments with the Academic Editor. 

Based on the reviews (attached below), we will probably accept this manuscript for publication, provided you satisfactorily address the remaining points raised by the reviewers. Please also make sure to address the following data and other policy-related requests:

a) We ask that you please take into account the concerns of Reviewer #2 and soften the claims on the nature of MgtE as an ion channel in your revised manuscript. Please re-phrase your wording in the discussion section (e.g. ‘the data is most consistent with the protein acting as an ion channel’). 

b) We would like you to consider a suggestion to improve the title as follows:

"The structure of MgtE in the absence of magnesium provides new insights into channel gating”

We expect to receive your revised manuscript within two weeks. 

- a cover letter that should detail your responses to any editorial requests, if applicable, and whether changes have been made to the reference list.

*Published Peer Review History*

*Early Version*

Sincerely,

Richard

Richard Hodge, PhD

Associate Editor, PLOS Biology

rhodge@plos.org

2) Deposition in a publicly available repository.

Fig. 1D-H; Fig. 2C; Fig. 4C; Fig. 5E; Fig. S2A; Fig. S5B, Fig. S8A-E, Fig. S10A, B and S12

*Please also ensure that figure legends in your manuscript include information on WHERE THE UNDERLYING DATA CAN BE FOUND where the underlying data can be found, and ensure your supplemental data file/s has a legend*

Please ensure that your Data Statement in the submission system accurately describes where your data can be found. In your revised submission, please delete the following sentence in the ‘Additional data availability information’ box:

‘Tick here if the URLs/accession numbers/DOIs will be available only after acceptance of the manuscript for publication so that we can ensure their inclusion before publication.’

* Please also make the structural data deposited in the PDB (6LBH) and EMD (EMDB-0869) publicly available.

Reviewer remarks:

Reviewer #1: The authors have address my concerns. I have no further comments on the manuscript.

Reviewer #2: Overall, I am satisfied with the revised version of the paper. I am still not convinced that MgtE is a channel, but on the balance of it, I think its more likely to be a channel than a secondary-active transporter. I think the authors have made their scientific arguments clearer and their data will no doubt contribute to a clearer function of this protein.

---

## [Editor Report · Decision Letter 3]

12 Apr 2021

Dear Dr Hattori,

On behalf of my colleagues and the Academic Editor, Raimund Dutzler, I am pleased to say that we can in principle offer to publish your Research Article "The structure of MgtE in the absence of magnesium provides new insights into channel gating" in PLOS Biology, provided you address any remaining formatting and reporting issues. These will be detailed in an email that will follow this letter and that you will usually receive within 2-3 business days, during which time no action is required from you. Please note that we will not be able to formally accept your manuscript and schedule it for publication until you have made the required changes.

PRESS

Thank you again for supporting Open Access publishing. We look forward to publishing your paper in PLOS Biology. 

Sincerely, 

Richard

Richard Hodge, PhD

Associate Editor, PLOS Biology

rhodge@plos.org

PLOS
